# A novel image segmentation method based on spatial autocorrelation identifies A-type potassium channel clusters in the thalamus

**Csaba Dávid[1,2], Kristóf Giber[1], Katalin Kerti-Szigeti[3,4], Mihály Köllő[3,5], Zoltan Nusser[3], Laszlo Acsady[1]***

[1]Lendület Laboratory of Thalamus Research, HUN-REN Institute of Experimental Medicine, Budapest, Hungary; [2]Department of Anatomy, Histology and Embryology, Semmelweis University, Budapest, Hungary; [3]Laboratory of Cellular Neurophysiology, HUN-REN Institute of Experimental Medicine, Budapest, Hungary; [4]Novarino Group, Institute of Science and Technology, Klosterneuburg, Austria; [5]Sensory Circuits and Neurotechnology Laboratory, Francis Crick Institute, London, United Kingdom

**\*For correspondence:**
acsady@koki.hu

**Competing interest:** The authors declare that no competing interests exist.

## eLife assessment

The manuscript introduces an **important** and innovative non-AI computational method for segmenting noisy grayscale images, with a particular focus on identifying immunostained potassium ion channel clusters. This method significantly enhances accuracy over basic threshold-based techniques while remaining user-friendly and accessible, even for researchers with limited computational backgrounds. The evidence supporting the method's efficacy is **convincing**. Its practical application and ease of use make it a tool that will benefit a wide range of laboratories.

**Abstract** Unsupervised segmentation in biological and non-biological images is only partially resolved. Segmentation either requires arbitrary thresholds or large teaching datasets. Here, we propose a spatial autocorrelation method based on Local Moran's *I* coefficient to differentiate signal, background, and noise in any type of image. The method, originally described for geoinformatics, does not require a predefined intensity threshold or teaching algorithm for image segmentation and allows quantitative comparison of samples obtained in different conditions. It utilizes relative intensity as well as spatial information of neighboring elements to select spatially contiguous groups of pixels. We demonstrate that Moran's method outperforms threshold-based method in both artificially generated as well as in natural images especially when background noise is substantial. This superior performance can be attributed to the exclusion of false positive pixels resulting from isolated, high intensity pixels in high noise conditions. To test the method's power in real situation, we used high power confocal images of the somatosensory thalamus immunostained for Kv4.2 and Kv4.3 (A-type) voltage-gated potassium channels in mice. Moran's method identified high-intensity Kv4.2 and Kv4.3 ion channel clusters in the thalamic neuropil. Spatial distribution of these clusters displayed strong correlation with large sensory axon terminals of subcortical origin. The unique association of the special presynaptic terminals and a postsynaptic voltage-gated ion channel cluster was confirmed with electron microscopy. These data demonstrate that Moran's method is a rapid, simple image segmentation method optimal for variable and high noise conditions.

## Introduction

Differentiation of signal from noise is a key step in any image segmentation process (*Pham et al., 2000*) and is based on the assumption that pixels constituting the image can be divided into spatially non-overlapping regions which are characterized by statistically similar intrinsic features (*Haralick and Shapiro, 1985*; *Pal and Pal, 1993*).

During the segmentation process, the signal can be defined as a spatially congruent population of pixels with gray values, which are statistically different from the neighboring pixels, that is, from the background (*Pham et al., 2000*; *Pal and Pal, 1993*). A theoretically optimal segmentation process should be independent of the person performing the analysis and should be based on the intrinsic properties of the image rather than manually defined values. It should consider not only the intensities but also the spatial arrangement of the picture elements (i.e., pixels).

The current segmentation methods are either based on hard or soft computing (*Sinha et al., 2020*). Though there is huge surge of soft computation methods based on fuzzy logic, deep learning methods neuronal networks and artificial intelligence (*Dey et al., 2018*; *Hemanth and Anitha, 2014*; *Devi et al., 2021*), image analysis based on binary, hard computation methods have their own advantages (*Jena et al., 2020*) and best results can be expected from the combination of the two (*Perfilieva et al., 2019*). Many hard computing image segmentation methods depend on thresholding (*Sinha et al., 2020*; *Sahoo et al., 1988*). Although more advanced threshold-based methods (TBMs) utilize local not global thresholds or multilevel thresholding (*Manikandan et al., 2014*; *Umaa Mageswari et al., 2013*), the threshold is still defined using the intrinsic properties of the image (e.g., as deviation from the mean intensity in the picture), selection of the degree of deviation is subjective and can yield different results (*Figure 1—figure supplement 1*). Furthermore, even in local thresholding there is no guarantee that spatially inhomogeneous noise reaching or approaching the intensity of the signal will not be included as signal.

Here, we propose a novel hard computing method for unsupervised image segmentation based on Local Moran's spatial correlation coefficient (Local Moran's *I*) (*Anselin et al., 2006*). The method was originally invented for geoinformatics and was used to study large-scale spatial correlation of geological (*Moran, 1948*; *Getis and Ord, 1992*), social, or medical events for example suicide rates among counties (*Helbich et al., 2015*) or in geomorphological studies (*Alvioli et al., 2016*). The method, as used here, is based on the statistical evaluation of intensities among neighboring pixels, estimating the probability whether a given pixel intensity arrangement is due to a random process or not. The proposed method allows for selecting spatially congruent pixels with statistically similar intensity (i.e., signal and background) and pixels with intensity distribution not significantly different from random (i.e., noise).

In this paper, we systematically compare the performance of TBM and Local Moran's method using images with different levels of noise. We show that, especially at high noise levels, Moran's method convincingly outperforms TBM on both artificial and natural images. Finally, we also test Moran's in a biological problem: clustering of voltage-gated ion channels in neuronal membranes. We show that the method can segment ion channel clusters, which are biologically relevant since their existence can be demonstrated using independent methods.

## Results

In order to demonstrate the way Local Moran's method works for image segmentation, we used high magnification fluorescent (i.e., dark field) (*Figure 1a–j*) as well as bright field (*Figure 1k–t*) light microscopic images showing immunoreactivity for Kv4.2 potassium channel subunit in the ventral posteromedial (VPM) nucleus of the mouse thalamus. In our approach, the unit of the analysis is the individual pixel of the image.

The method is based on testing the null hypothesis that pixel intensity distribution in the image is random, which is analyzed with the aid of convolution matrices. The results are tested by Monte Carlo simulation to identify whether the pixel intensity distribution is non-random in an image (*Figure 1*). In dark field images, contiguous clustered pixels with similarly high intensities represent object (red in *Figure 1j*), clustered pixels with low intensities indicate no object (e.g., lumen of a blood vessel) (blue in *Figure 1j*). Group of pixels with random intensity value distribution (which do not reach significance)

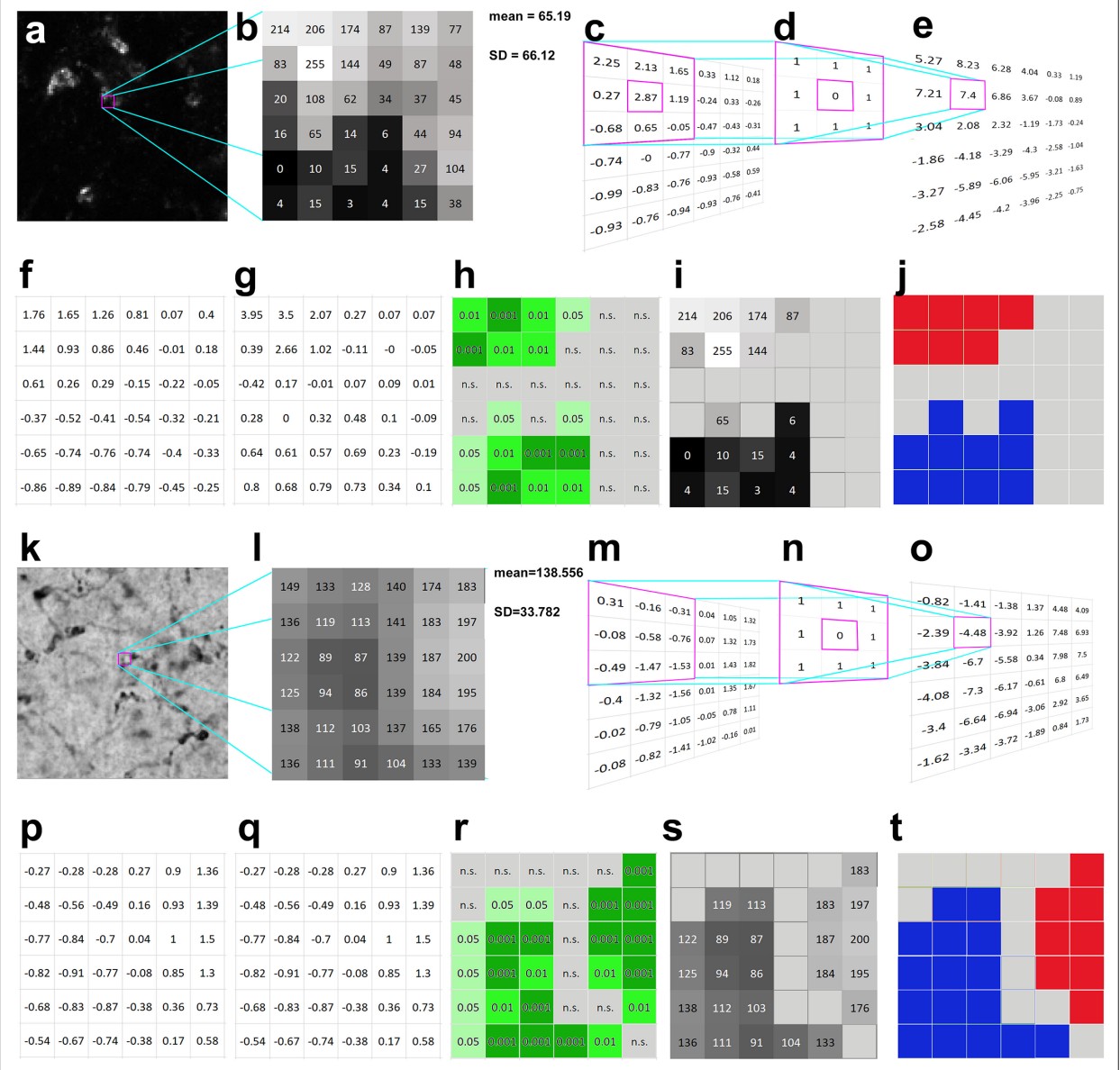

**Figure 1.** Calculation of Local Moran's $I$ in darkfield (**a–j**) and brightfield (**k–t**) light micrographs. (**a**) Large magnification fluorescent immunostaining for Kv4.2 in the ventral posteromedial (VPM) nucleus. (**b**) A 6 × 6 pixel segment of the image shown in a. The squares represent the pixels with grayscale values. (**c**) Normalized intensity values ($z_i$). The purple framed area demonstrates the convolution step of calculations (see also methods steps 1–3). (**d**) A first-order convolution matrix (weight matrix, $w_{ij}$) was used for this calculation. (**e**) Weighted intensity values ($w_{ij}z_j$). (**f**) Lagged gray values. (**g**) Local Moran's $I$ ($I_i$) values. (**h**) Pseudosignificance values for each pixel. (**i**) Masking the non-significant pixels (middle gray tone without numbers) resulted in two groups of pixels, where the intensity values significantly differ from a random distribution. (**j**) Considering the intensity values, the upper pixel group is interpreted as an object (red), the lower one as background (blue), and the gray field between them represent an area where the pixel intensities could come from a random distribution (interpreted as noise). (**k–t**) The same as a–j, but the calculations are shown on a bright field image. In this case, the low-intensity pixels representing the object are labeled with blue (**t**) high-intensity pixels (background) with red. Scale bars: (**a**) 5 μm, (**k**) 10 μm.

The online version of this article includes the following figure supplement(s) for figure 1:

**Figure supplement 1.** Effect of threshold level on object detection.

represent noise (gray in *Figure 1j*). In bright field images, the picture is reversed, contiguous low- and high-intensity pixels represent the object and no object, respectively (blue and red in *Figure 1t*).

Local Moran's $I$ differentiates between object, no object, and noise in the following way: the method first normalizes each pixel intensity value (*Figure 1c, m*) (see Methods). Next, it defines a weight matrix in the local neighborhood (3 × 3 in this particular case) for each pixel (*Figure 1d, n*), where the

pixel in question receives zero weight. Then it calculates a weighted intensity value for each pixel in the image using the summed values of the neighboring pixels (*Figure 1e, o*). Immediate neighbors of each pixel are assigned with the weight of 1. Next, it creates lagged gray values by dividing the matrix values with the number of neighboring pixels (*Figure 1f, p*). This was called 'lagged' in the original description (*Anselin, 1995*), since the distance (lag) was considered in the calculation of this particular value (which was 1 in our case). Finally, it generates the Local Moran's *I* value for each pixel by multiplying the lagged gray values (*Figure 1f, p*) with the normalized values (*Figure 1c, m*).

The resulting Local Moran's *I* value (*Figure 1g, q*) will represent how the intensity of the pixel and its neighbors differs from the distribution of intensities of all pixels in the image (clusteredness). We can test the salience of this value if we repeat the same procedure with randomly selected neighboring pixels. For each pixel, the method determines the probability (pseudosignificance) of obtaining this or higher Local Moran's *I* value using Monte Carlo simulations of the intensity values of all pixels in the image (*Figure 1h, r*). Significant pixels (p <0 .05) with high intensities (in case of dark field images) or low intensities (in case of bright filed images) will be considered as object. These steps ensure that signal is not determined by an arbitrarily defined threshold, but by the combination of spatial arrangement and pixel intensities. As a consequence, in this method consideration of each pixel as signal will not depend purely on the absolute intensity value of the pixel but on the intensity values of the neighboring pixels and the statistical probability that this spatial arrangement can arise (or not) randomly.

## Comparison of TBM and Moran's method on artificial images

We compared the performance of Moran's and TBM on artificial binary objects with various levels of Gaussian noise (*Figure 2a*). Since in this case we defined the objects (the ground truth), true positive rates (TPR = true positive pixels/number of object pixels) and false positive rates (FPR = false positive pixels/number of background pixels) could be directly determined.

Since Moran's method can be performed with convolution matrices of different sizes (i.e., Moran's order, see Methods; *Moran, 1948*) we first determined the optimal matrix size for different object sizes at variable noise levels on circular artificial objects (*Figure 2—figure supplement 1a*). First-order matrix contains the immediate neighboring pixels of the central pixel (3 × 3), the second-order matrix contains all the immediate neighbors of the first-order pixels (5 × 5), and so on. We found that optimal Moran's order was independent of the noise level, and it was 0.25 times (range: 0.238–0.253) of the object size (*Figure 2—figure supplement 1b*). Thus, for these measures, we selected Moran's order 4. For the TBM, the threshold was set to 50% of the intensity level of the objects.

At zero noise, both methods had 1 TPR and 0 FPR values (*Figure 2a–c*). Increasing noise levels resulted in a parallel decrease in TRP values. However, in case of the TBM FPR increased dramatically (cyan dots in *Figure 2b*) due to the higher fluctuation of pixel values (*Figure 2d*), which indicates a highly unreliable exclusion of false positive pixels (*Figure 2e*, circles), while the FPR value in case of Moran's method remained nearly constant (*Figure 2e*, triangles) with increasing noise. In case of TBM FPR increase was the consequence of that random noise frequently increased the pixel intensity values of the background above the threshold level (*Figure 2*), which were detected as positive pixels.

Next, we compared the two methods in a larger sample of images (*n* = 100) which contained 64 randomly distributed objects of different sizes. The diameter of the objects varied between 4 and 64 pixels in each image (*Figure 2a*). Here, we also examined the effect of Moran's order on the quality of detection (*Figure 2—figure supplement 2*). Similar to same-sized objects in images containing variable-sized objects at 0 noise level, both methods selected the objects equally well (TRP = 1, SD = 0, FPR = 0, SD = 0, *n* = 100 images). However, increasing the noise level resulted again in a pronounced increase in FPR in the case of TBM (*Figure 2b*) whereas using Moran's, this effect was evident only at the edges of detected objects and FPR just marginally increased even at very high noise levels (*Figure 2c*). Huge increase in false positive pixels in TBM with increasing noise resulted in significantly lower FPR values compared to that of Moran's (*Figure 2e, f*, FPR values noise 80: threshold, 0.13333 ± 0.0009, Moran's: 0.0085 ± 0.0003, p < 5.4*10$^{-215}$; noise 160: threshold, 0.3866 ± 0.0012, Moran, 0.0055 ± 0.0005, p < 2.1*10$^{-244}$). TPR of Moran's was also significantly better at noise 80 (*Figure 2e, f*, TPR values noise 80: threshold, 0.9116 ± 0.0023; Moran's, 0.9674 ± 0.0025, p < 2.7*10$^{-130}$). The TPR of TBM was marginally better at noise 160 (*Figure 2e, f*, noise 160: threshold, 0.6799 ± 0.0041; Moran's: 0.6609 ± 0.0060, p < 5.04*10$^{-69}$), but the difference in FPR values was so large between the two methods that it far outweighed this effect. We defined the quality of detection

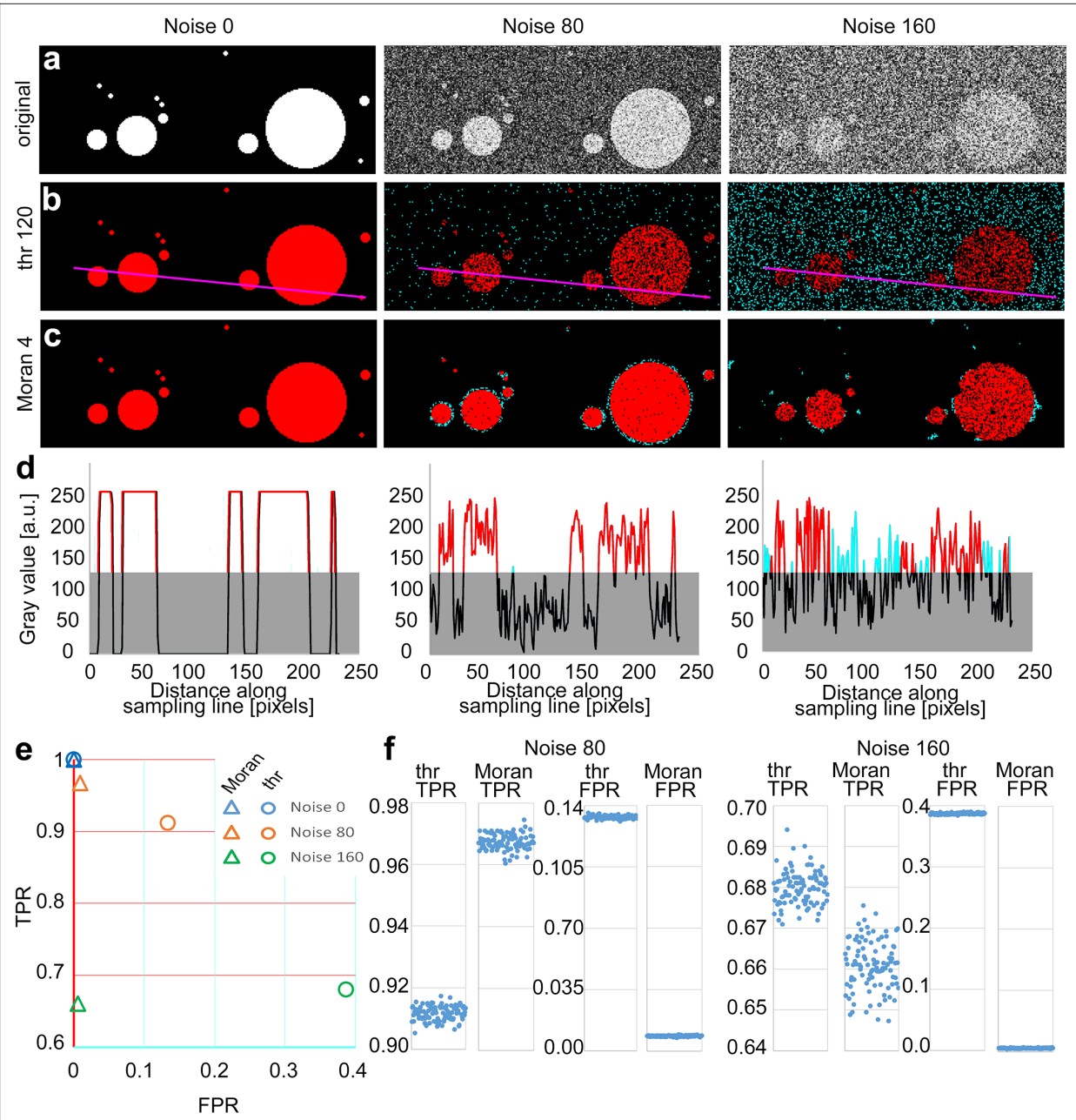

**Figure 2.** Comparison of threshold-based method (TBM) and Moran's image selection on artificial objects. (**a**) A series of artificial objects with different diameters (white circle on black background) was defined, and varying levels of noise were added (first row). The objects are segmented using the threshold (**b**) and Moran's *I* method (**c**). Red, true positive (TP) pixels; cyan, false positive (FP); black, true negative; black, true and false negative. (**d**) Pixel intensity values along the sampling lines at the three noise levels. Gray rectangles indicate the threshold level. Portions of the graph's line are color-coded (red: true positive, cyan: false positive, black: negative). (**e, f**) Average true positive rate (TPR) and false positive rate (FPR) values of Moran's and TBM at different noise levels from 100 random samples. Individual data points are shown separately at a different scale (**f**).

The online version of this article includes the following source data and figure supplement(s) for figure 2:

**Source data 1.**

**Figure supplement 1.** Defining optimal Moran's matrix.

**Figure supplement 2.** The effect of Moran's order on the quality of detection using variable object sizes and different noise levels.

**Figure supplement 3.** Comparison of the vector length at various noise levels (20–240) from the theoretical best (0, 1 point in the true positive rate [TPR]/false positive rate [FPR] space) to the closest point of the TPR/FPR curve in case of the Tris-buffered saline (TBS) and Moran's methods using images containing randomly placed artificial objects with diameter ranging from 4 to 64 pixels.

as a vector in the FPR/TPR space starting from the FPR = 0, TPR = 1 point. Quality of detection was significantly better with Moran's method at both noise levels in a population of 100 randomly generated images (*Figure 2e*, FPR/TPR vector noise 80: threshold, $0.1600 \pm 2.2*10^{-6}$; Moran's: $0.0337 \pm 5.8*10^{-6}$, $p < 5*10^{-169}$; noise 160: threshold, $0.5019 \pm 7.5*10^{-6}$, Moran, $0.3391 \pm 3.7*10^{-5}$, $p < 10^{-156}$).

Finally, to gain insight into the performance of the two detection methods in a larger parameter space, we repeated the comparison using 13 Moran's order, 13 threshold levels, and 13 noise levels (*n* = 100 images for each condition). We found that with the exception of the lowest noise levels (0–30) Moran's method outperformed TBM at every noise levels irrespective of the threshold mostly independent of the exact Moran's order used (range 4–13; *Figure 2—figure supplement 3*). These data demonstrate that, especially in high noise situations, Moran's method successfully eliminates background noise and selects objects with extremely low FPR.

## Comparison of TBM and Moran's methods on natural images

Next, we tested the performance of the two methods on grayscale images of natural objects (*Figure 3a*) where, by definition, the ground truth is not known, and therefore the comparison of the two methods to a common reference is not possible. To circumvent this problem, we chose a high magnification confocal image of a nerve tissue immunostained with vesicular glutamate transporter (vGluT2), a ubiquitous marker of axon terminals with subcortical origin in the thalamus (*Lavallée et al., 2005*; *Graziano et al., 2008*; *Rovó et al., 2012*). We prepared images with three different added noise levels and asked 23 expert human observers, who were blind to the nature of the image and the nature of the investigation, to delineate objects (*Figure 3a, b* and *Figure 3—figure supplement 1*). Observers always started delineation in the images with the highest noise level. Next, we generated a heat map and labeled the number of experts selecting each individual pixel. For the purpose of this study, we defined ground truth as pixels which were selected by at least 12 experts (*Figure 3—figure supplement 1*) and compared the performance of the two methods to this common reference (*Figure 3c–e*). The size of the objects (*n* = 36) in the image varied between 1 and 303 pixels corresponding to 0.043 and 13.003 μm$^2$, respectively. For a systematic comparison of the two image segmentation methods, we analyzed the data using different threshold levels and matrix sizes (*Figure 3—figure supplement 2*) at three noise levels (0, 80, and 160).

Similar to artificial objects, Moran's method outperformed TBM at every noise level. The FPR values remained low with Moran's method even at high noise levels at every Moran's order tested (*Figure 3f*). Thus, even though at the best threshold levels, the TPR values were comparable between the two methods, lower FPR values resulted in a better quality of detection (shorter vector) in the TPR/FPR space (*Figure 3f*).

It is important to emphasize that in the case of TBM at increased noise levels, false positive pixels were homogeneously distributed in the image due to the increased background noise (*Figure 3c*). In contrast, with Moran's, most false positive pixels surrounded the human-based selection areas (*Figure 3d*). This is largely the consequence of that Moran's method detected the out of focus pixels as well, which were routinely not selected by the human experts.

These data demonstrate that when natural images were analyzed, the quality of detection was better with Moran's method in every condition tested. The main strength of Moran's was that it did not include pixels of random noise, like TBM.

## Application of Moran's method for a biological problem

Next, we tested Moran's method to address a biologically relevant problem, the clustering of Kv4.2/4.3 subunits of the voltage-gated K$^+$ channels in thalamocortical neurons. Voltage-gated ion channels are either distributed homogeneously, along a gradient (*Lorincz and Nusser, 1979*; *Kerti et al., 2012*) or form high-density clusters in neuronal membranes (*Alonso and Widmer, 1997*; *Kollo et al., 2006*; *Kollo et al., 2008*), endowing neurons with unique electrical properties (*Chen et al., 2006*; *Hoffman et al., 1997*; *Johnston et al., 2003*; *Losonczy et al., 2008*). However, the exact subcellular distribution of these subunits and the logic of their possible clustering in thalamic cells are unknown. Therefore, we tested whether Kv4.2/4.3 ion channel clusters, detected by Moran's *I* method, form biologically relevant membrane protein aggregations.

Immunostainings for the Kv4.2 and Kv4.3 subunits displayed homogenous, diffuse labeling in the major somatosensory nucleus of the thalamus (VPM). At high magnifications, however, it became

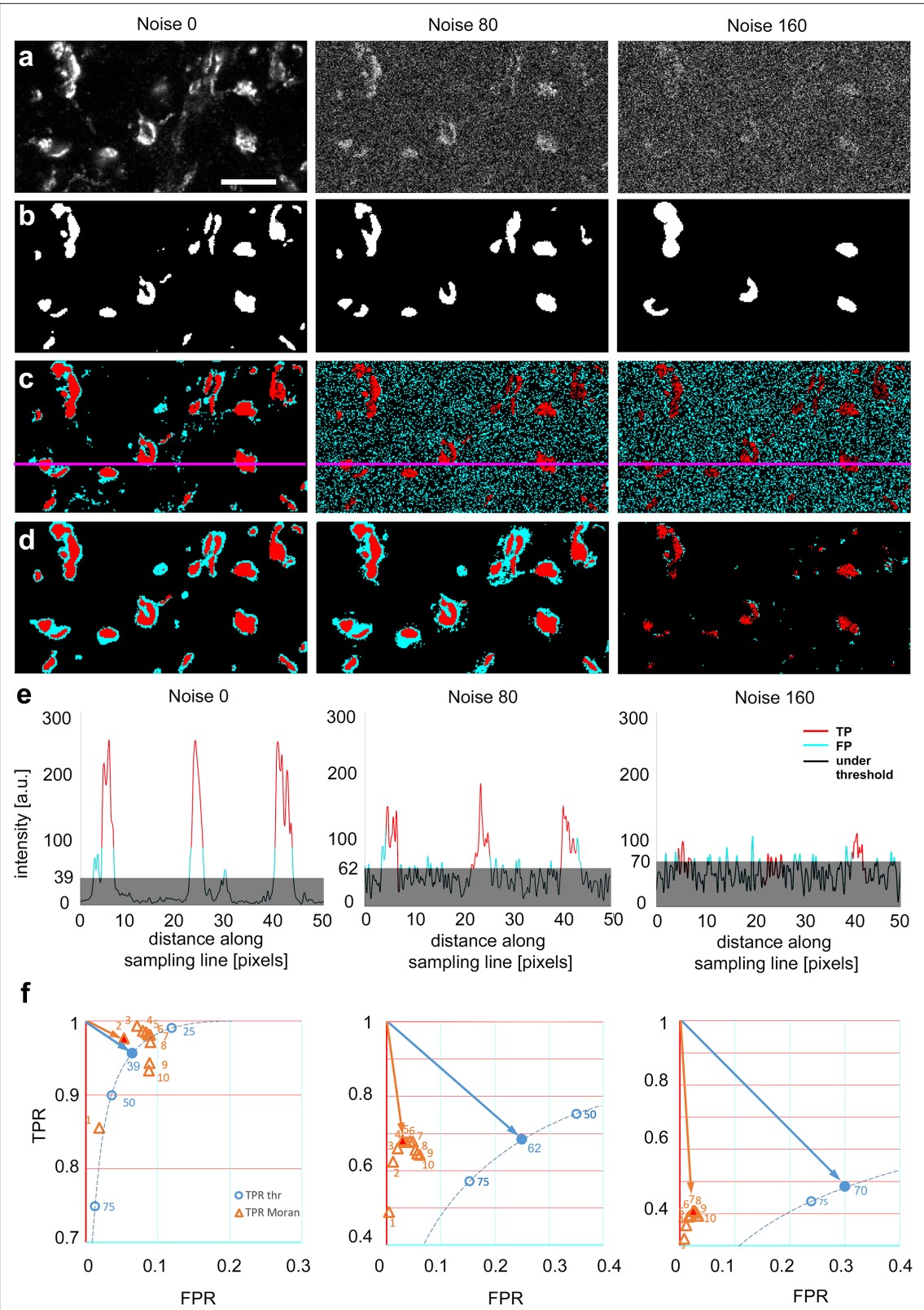

**Figure 3.** Comparison of threshold-based, and Moran's image selection on natural objects. (**a**) High-resolution confocal image (single optical plane) of Kv4.2 immunostaining in the VB complex of the mouse thalamus at three added noise levels (0, 80, and 160). (**b**) Majority projection of segmentation of the same images by expert human observers (*n* = 23, see methods). Pixels selected by the majority of observers are defined as ground truth. (**c**) Threshold-based segmentation with the best true positive rate [TPR]/false positive rate [FPR] values. (**d**) Moran's based segmentation using matrix

*Figure 3 continued on next page*

*Figure 3 continued*

size 4. (**e**) Pixel intensity values along the sampling lines at the three noise levels. Gray areas indicate threshold levels. Portions of the graph's line are color-coded (red: true positive, cyan: false positive, black: negative). (**f**) TPR and FPR values of Moran's and Tris-buffered saline (TBS) at different noise levels. The threshold curve (dotted, blue line) is computed using 254 threshold levels, some of which are labeled (blue circles). Filled blue circle, optimal threshold value. Orange triangle, TPR/FPR values of Moran's based segmentation using different matrix sizes. The size of the matrices is shown by numbers. Red filled triangle, optimal Moran's order. Orange and blue arrows indicate the distance of the optimal points from the theoretical best point (maximal TPR, zero FPR). Scale bar: (a) 10 μm.

The online version of this article includes the following source data and figure supplement(s) for figure 3:

**Source data 1.**

**Figure supplement 1.** Definition of ground truth on natural images.

**Figure supplement 2.** Effect of convolution matrix size on the visual results.

visually evident that Kv4 subunits also form strongly immunopositive clusters (2–10 μm; *Figure 4a*). To study the precise subcellular distribution of Kv4 subunits, we first examined the Kv4.2 subunit at the electron microscopic level using preembedding immunogold reactions. This approach confirmed the exclusive dendritic localization of this subunit (*n* = 123 profiles). As suggested by the light microscopic observations, large caliber dendrites of relay cells displayed a high density of immunogold particles for the Kv4.2 subunit (*n* = 9 animals, *Figure 4b, c*). Quantification of silver particles along the dendritic membranes (*n* = 3 animals) demonstrated more than three times higher density on thick (minor diameter over 1.2 μm) dendrites, preferentially innervated by large excitatory terminals containing round vesicles (RL terminals) compared to thin (minor diameter below 1.2 μm) dendrites which received very little of this type of inputs (*Figure 4d*).

To exclude the possibility that the uneven (i.e., clustered) immunogold labeling is the consequence of uneven accessibility of the antibodies to the epitopes in thick tissue and to better visualize protein aggregates at high resolution, we performed another electron microscopic experiment using the diffusion-free freeze-fracture replica immunogold labeling technique (*Masugi-Tokita and Shigemoto, 2007*; *Figure 4e–h*). Examination of replicas of the VPM immunolabeled for the Kv4.2 or the Kv4.3 subunits revealed gold particles associated with the protoplasmic face of dendritic segments, consistent with the intracellular location of the epitopes. On large diameter dendrites, we observed uneven distribution of gold particles (*Figure 4e*) consistent with our observations using confocal microscopy and preembedding immunogold localization. Quantitative evaluation of the reactions in one rat demonstrated that the density of gold particles for the Kv4.3 subunit was twice as high in large, compared to small diameter dendrites (12.6 ± 8.0 gold/μm², *n* = 18 vs 6.0 ± 3.6 gold/μm², *n* = 18). A similar difference was found for the Kv4.2 subunit, for which the density of gold particles was 22.8 ± 8.8 gold/μm² (*n* = 9) in large diameter proximal dendrites, whereas it was 9.3 ± 3.2 gold/μm² (*n* = 6) in small, distal dendrites. Dendritic appendages were also decorated with gold particles (*Figure 4h*).

These data together demonstrate that Kv4.2 and Kv4.3 subunits display weak distal dendritic labeling, but the proximal, large diameter dendrites contain high densities of these ion channel subunits in VPM.

Next, we investigated whether the Moran's method as described above can delineate Kv4.2 subunit clusters and quantitatively characterize them in the VPM (*Figure 5*). Moran's method clearly segmented immunopositive and negative regions (*Figure 5a–c*). The immunopositive regions displayed large variability in size intensity and contained diffuse labeling and high-intensity clusters as suggested by electron microscopic investigation. To test to what extent Kv4.2 clusters detected by Moran's method represent ion channel clusters identified on the proximal dendrites of VPM cells at the electron microscopic level (*Figure 4*), we visualized large excitatory axon terminals known to be associated with the proximal dendrites of VPM cells (*Sherman and Guillery, 2006*). These subcortical excitatory inputs were labeled with vesicular glutamate transporter type 2 (vGluT2, *Figure 5d*; *Herzog et al., 2001*; *Fremeau et al., 2001*). We selected the proximal dendrites by performing segmentation using Moran's method in both channels of the Kv4.2/vGluT2 double immunofluorescent images. The data showed that large, high-density clusters of Kv4.2 ion channels were indeed associated with vGluT2-immunopositive terminals (*Figure 5e*, cyan arrows), whereas the smaller and less intense ones were not (*Figure 5f*, yellow arrows). Immunopositive clusters generally had highly variable sizes (5.59 ± 13.26 μm²) and intensities (745 ± 260 a.u.). The large majority of the non-colocalizing Kv clusters were below 1 μm² and faintly stained (*Figure 5g*) whereas more than the half of the colocalized

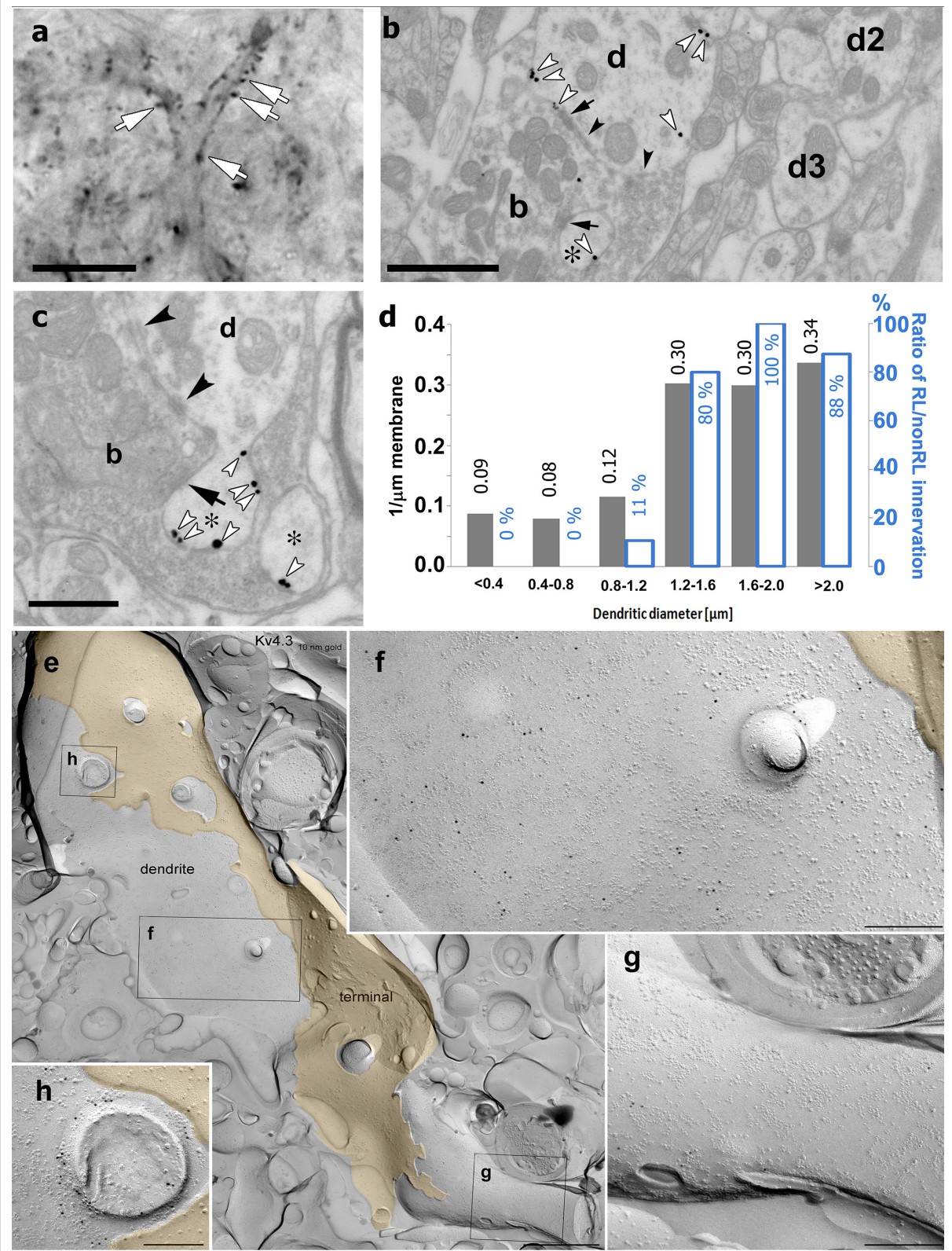

**Figure 4.** High-resolution localization of Kv4.2 subunit in the ventral posteromedial (VPM) nucleus of the thalamus. (**a**) High-resolution light micrograph of a VPM neuron immunolabeled for the Kv4.2 subunit displaying intense immunopositive puncta (white arrows) along its proximal dendrites. (**b**) An electron micrograph showing Kv4.2 immunolabeling in the VPM using the preembedding immunogold technique. A proximal dendrite, identified by its large diameter (**d**), has more silver intensified gold particles (white arrowheads) than thin dendrites (**d2, d3**). Arrows, synapses; arrowheads, puncta

*Figure 4 continued on next page*

*Figure 4 continued*

adhaerentia. (**c**) Accumulation of gold particles in small dendritic appendages (asterisks). (**d**) Quantitative analysis revealed an approximately threefold higher density of gold particles along the membrane in large (>1.2 µm) compared to small dendrites (left *y*-axis, gray bars). The histogram also shows the ratio of large excitatory terminals with round vesicles (RL) innervating each dendritic category (right *y*-axis, white bars). (**e**) Freeze-fracture replica labeling of the protoplasmic-face membrane segment of a large thalamocortical cell dendrite in VPM is shown with intense immunogold labeling for the Kv4.3 subunit. The dendrite is surrounded by a large axon terminal (yellow). Boxed areas are shown at higher magnifications in (**f–h**). (**f–h**) Membrane area in the vicinity of the axon terminal contains a high density of gold particles (**f**), whereas a neighboring area (**g**) has very few gold particles, indicating an inhomogeneous distribution of the Kv4.3 subunit. Note the intense immunogold labeling around a cross-fractured dendritic appendage (**h**). Black arrows, synapses; black arrowheads, puncta adherentia. Scale bars: (**a**) 10 µm, (**b**) 1 µm, (**c**) 0.5 µm, (**e**) 1 µm, (**f–h**) 200 nm.

The online version of this article includes the following source data for figure 4:

**Source data 1.**

Kv4.2 structures were above 5 µm$^2$ and more intensely labeled (*Figure 5g*). Kv4.2 clusters associated with vGluT2-positive terminals were significantly larger (12.21 ± 18.63 µm$^2$) and more intense (894 ± 280 a.u.) than Kv4.2 regions outside the vGluT2 terminals (average size: 0.90 ± 1.31 µm$^2$, average intensity: 639 ± 182 a.u.) indicating that Moran's *I* method could delineate functionally relevant ion channel clusters associated with a specific synaptic input (*Figure 5h*).

To verify the association of afferents and ion channel clusters, we performed double immunostaining for Kv4.2 and vGluT2 at the electron microscopic level (*Figure 5i, j*). Indeed, membrane-associated silver intensified gold particles for the Kv4.2 subunit were associated with membrane surfaces covered by vGluT2-positive terminals, as demonstrated by the Moran's method at the light microscopic level. Membrane segments contacted by vGluT2 boutons showed more than four times higher gold particle density (37.0 particles/100 µm membrane profile, total measured length: 601 µm) compared to surrounding membrane surfaces that were not contacted by vGluT2-positive axons (8.9 particles/100 µm, total measured length: 394 µm). These data reveal a novel association of voltage-gated ion channel clusters on the postsynaptic membrane surfaces with a specialized presynaptic terminal.

## Discussion

In this study, we demonstrate that Local Moran's method is suitable for image segmentation and outperforms TBM in both artificial and natural images, especially when the noise is substantial. We used this method to successfully delineate and quantify biologically relevant membrane protein clusters. Since Moran's selections are based on the relative intensity values of neighboring pixels, it can select contiguous points as objects, unlike TBM. TBM does not use spatial information of the pixels. Therefore, it selects random noise as well, provided it reaches the threshold. Unlike in TBM, selection of the signal will not depend on the absolute intensity of its pixels in Moran's method since the image processing starts with a normalization step. This way, it uses inherent properties of the image to select clusters. This will allow the systematic, quantitative comparison of images where sample preparation and image acquisition cannot be standardized, such as in most histological samples.

Local Moran's method has not been used for image segmentation so far. Its global version, however, has already been utilized in a wide range of biomedical image analyses. Global Moran's method assigns a value to the whole image and describes whether pixel distribution is random or not, but does not segment pixels to object, no object and noise. Global Moran's *I* were used to examine muscle and fat tissue on three-dimensional (3D) MRI (Magnetic Resonant Images) images (*Santago et al., 2016*) or to measure anisotropy in ultrasound imaging (*Ahn and Kaptchuk, 2011*). In addition, Global Moran's method was also used to analyze the homogeneity of the temporal information in Ca$^{2+}$ signals in periodontal fibroblasts (*Ei Hsu Hlaing et al., 2019*) or to detect correlated neuronal activity (*Schmal et al., 2017*). Based on these, the utilization of Local Moran's method, as described here, will not be restricted to segmenting objects on two-dimensional images but can be extended for 3D image segmentation and analyzing temporal signals.

The strongest argument to use Local Moran's *I* instead of threshold-based segmentation is the effective elimination of false positive pixels from the background. Although the intensity of noisy pixels can reach or exceed that of the objects, white noise does not cause a pattern. Detection of the lack of spatial correlation is a powerful tool to ignore false positive salient pixels in the background.

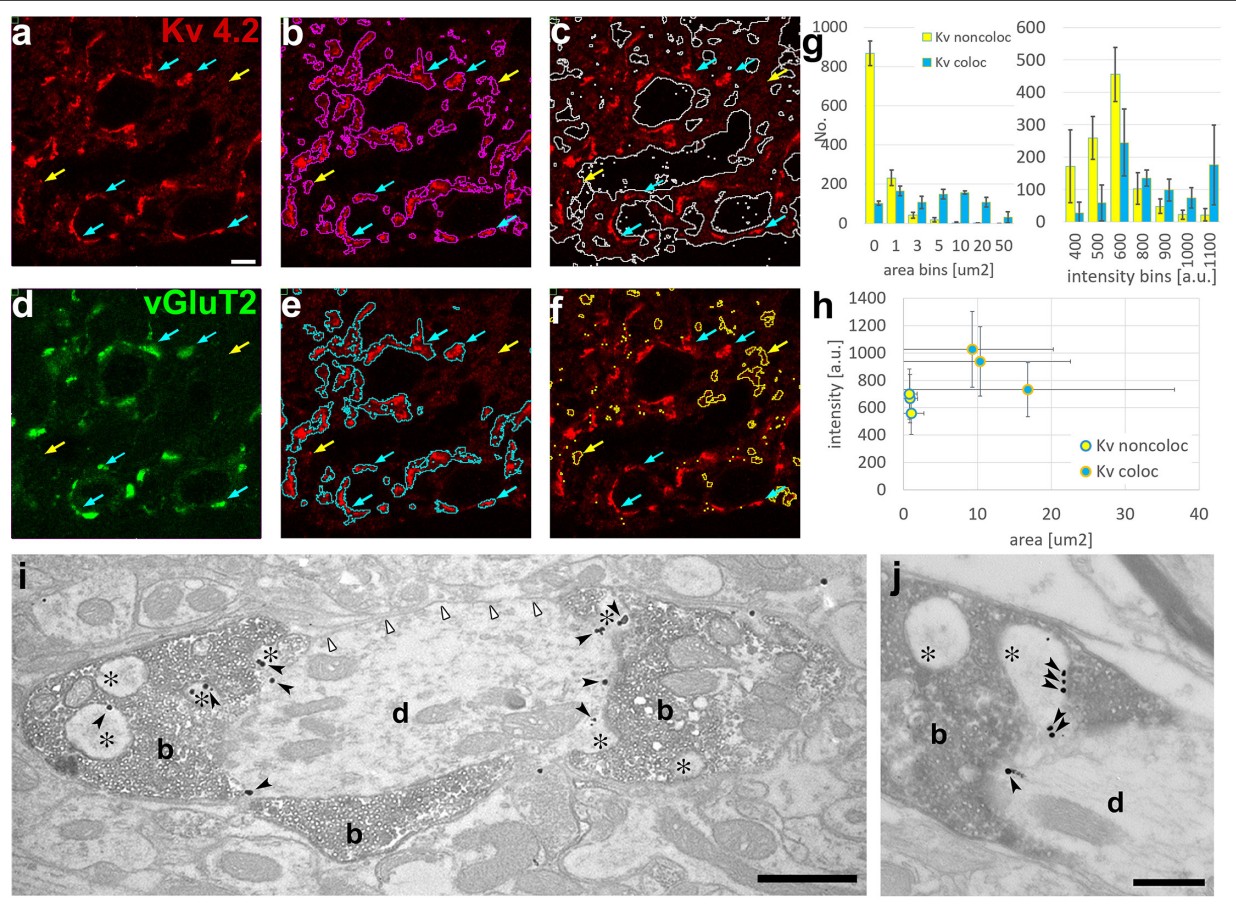

**Figure 5.** Delineation of functionally relevant Kv4.2 ion channel clusters by Moran's method. (**a**) High magnification confocal image of Kv4.2 immunostaining in the ventral posteromedial (VPM) nucleus of mouse thalamus. (**b**) Segmentation of Kv4.2 immunopositive objects by Moran's (magenta outlines). (**c**) Segmentation of immunonegative areas by Moran's (white outlines). (**d**) Confocal image of vGluT2 immunostaining (green channel) of the same section shown in (**a**). (**e**) Kv4.2 ion channel clusters which overlap with vGluT2-positive terminals (cyan outlines). (**f**) Kv4.2 ion channel clusters which do not overlap with vGluT2-positive terminals (yellow outlines). Blue arrows, Kv4.2 clusters overlapping with vGluT2 terminals; yellow arrows, Kv4.2 clusters non-overlapping with vGluT2 terminals (holds for a–f). (**g**) Histograms of the colocalizing and non-colacalizing Kv 4.2 immunopositive boutons pooled from three animals segmented by Moran's method and binned by area (left) and intensity (right). Yellow bars, Kv4.2 objects non-overlapping with vGluT2; cyan bars, Kv4.2 objects overlapping with vGluT2. (**h**) Scatterplot of area vs staining intensity from three different animals. Yellow dots, Kv4.2 objects non-overlapping with vGluT2; cyan dots, Kv4.2 objects overlapping with vGluT2. (**i, j**) Silver particles (black arrowheads) indicating Kv4.2 immunoreactivity are localized on dendritic appendages (asterisks) and dendritic shafts (**d**) contacted by vGluT2-positive terminals (b, diaminobenzidine [DAB] positive) in the VPM. The plasma membrane of the same dendrites that are not contacted by vGluT2-positive terminals (white arrowheads) contains very few particles for the Kv4.2 subunit. Scale bars: (**a–f**) 5 μm, (**i**) 1 μm, (**j**) 0.5 μm.

The online version of this article includes the following source data for figure 5:

**Source data 1.**

As indicated above, Moran's method works without user-dependent intensity values. However, the size of the convolution matrix needs to be set by the user. In this study, we examined the effect of matrix size in a wide range of applications (***Figure 3f***, ***Figure 3—figure supplements 1, 2***). We determined that selection of the signal is relatively insensitive of the exact matrix size within a certain range (Moran's order 3–8), especially in noisy conditions. A priori knowledge of the expected object size helps to define the best matrix size since we found a linear correlation between object size and Moran's order for optimal image segmentation (***Figure 2—figure supplement 1***).

We found that Local Moran's method can delineate and quantitatively characterize Kv4.2 ion channel clusters in the somatosensory thalamus formed in the proximity of large vGluT2-positive afferent nerve terminals. The selective association of a given presynaptic terminal type and the post-synaptic accumulation of ion channel clusters were verified by independent electron microscopic

methods. The functional relevance of this unique association remains to be determined: it may participate in timing the precision of sensory integration, but recently the non-synaptic role of voltage-gated ion channel accumulation has also been proposed (*Cserép et al., 2020*).

# Methods
## Animals
Six adult male C57Bl/6 mice were perfused to examine the Kv4.2 expression. All animals were bred and kept in the Animal Facility of the Institute of Experimental Medicine, Budapest, Hungary. All procedures with mice were approved by the Animal Welfare Committee of the Institute of Experimental Medicine, Budapest, conformed to guidelines established by the European Community's Council Directive of November 24, 1986 (86/609/EEC). Experiments were authorized by the National Animal Research Authorities of Hungary (PE/EA/877-7/2020).

The animals were deeply anesthetized with a mixture of ketamin, xylazine (Produlab Pharma BV, Raamsdonksveer, The Netherlands), and promethazine (Pipolphen, EGIS, Budapest, Hungary; 2, 10, and 5 mg/kg body weight, respectively) and perfused through the heart using a fixative containing 2% or 4% paraformaldehyde (TAAB, Berkshire, UK) and 15% saturated picric acid (Spektrum-3D Kft, Debrecen, Hungary) dissolved in 0.1 M phosphate buffer (PB).

## Immunohistochemistry, light microscopy
Coronal sections (50 μm) were cut from the thalamus, and thoroughly rinsed in 0.1 M PB followed by 0.1 M Tris-buffered saline (TBS, pH = 7.4). For the investigation of the distribution of A-type $K^+$ channel subunits in different nuclei of the thalamus, the following antibodies were used; mouse anti-Kv4.2 (from NeuroMab Inc, UC Davis, CA; 1:300, codes 75-016 and 75-017); rabbit anti-Kv4.2 (from Alomone labs, Jerusalem, Israel, 1:500, code APC-023 and APC-017). This was followed by the incubation with biotinylated horse anti-mouse or goat anti-rabbit antibodies (bHAM or bGAR; Vector Laboratories, 1:300; in TBS; 3 hr, codes BA-2000 and BA-1000, respectively) and by avidin-biotinylated horseradish peroxidase complex (ABC, Vector, 1:300; 90 min, code PK-6200). The immunoreactions were visualized using diaminobenzidine (DAB, Sigma) or nickel intensified DAB (DAB-Ni) as chromogens. Sections were treated with $OsO_4$ (1% in 0.1 M PB; 40 min, EMS, Munich, Germany), dehydrated in ethanol and propylene oxide, and embedded in Durcupan (ACM, Fluka, Buchs, Switzerland).

For the visualization of the vesicular glutamate transporter 2 (vGlut2) and the Kv4.2 subunit, fluorescent double labeling experiments were carried out. The sections were incubated with a mixture of mouse anti-Kv4.2 (1:300) and rabbit anti-vGlut2 (1:1500; Synaptic Systems, Göttingen, Germany, code 135 402) or rabbit anti-Kv4.2 and mouse anti-vGlut2 (1:3000, Millipore, Temecula, CA, code MAB 5504). This was followed by a mixture of Alexa488-conjugated goat anti-rabbit (A488-GAR; 1:500; 2.5 hr; Molecular Probes, Leiden, The Netherlands, code A-11001) and Alexa594-conjugated goat anti-mouse (A594-GAM; 1:500; 2.5 hr; Molecular Probes, Leiden, The Netherlands, code A-11012) IgGs. In some experiments Cy3-conjugated goat anti-mouse (1:500 Jackson ImmunoResearch Europe Ltd, Suffolk UK, code 111-165-144) IgGs were used instead of Alexa594. After the reactions, the sections were mounted on glass slides in Vectashield (Vector Laboratories, Burlingame, CA) and were examined using a Zeiss Axioplan2 microscope (wavelengths for filter set absorption and emission in nm: 365 bandpass/420–460; 450–490/512–542; 540–546/578–643). Digital micrographs were taken with a digital camera (Olympus, DP70, Tokyo, Japan). Z image stacks were also acquired with a confocal laser scanning microscope (Olympus FV1000 or Nikon ECLIPSE Ni-E).

## Freeze-fracture replica labeling
80 μm coronal sections were cut from the forebrain of a perfusion fixed Wistar rat (2% paraformaldehyde and 15% vol/vol picric acid in 0.1 M BP) and small blocks from the VPM were cut out for rapid freezing with a high pressure freezing machine (HPM100, Leica Microsystems) and were fractured and coated as described earlier (*Kerti et al., 2012*).

Replicas were digested in a solution containing 2.5% sodium dodecil sulphate and 20% sucrose in TBS (pH = 8.3) at 80°C for 18 hr Following several washes in TBS containing 0.05% bovine serum albumin (BSA), replicas were blocked in TBS containing 5% BSA for 1 hr, then incubated overnight in blocking solution containing either a rabbit anti-Kv4.3 (1:500; Chemicon/Millipor, code AB5194) or a

rabbit anti-Kv4.2 IgG (1:500; Alomone Labs, code APC-023). After this, the reactions were visualized with goat anti-rabbit IgGs coupled to 10 nm gold particles (1:100; British Biocell International).

## Calculation and properties of Moran's *I*

The code for the analysis is available online as a user-friendly MATLAB script at: https://github.com/dcsabaCD225/Moran_Matlab/blob/main/moran_local.m.

In details, methods for detecting spatial inhomogeneities (clustering) in geostatistics were described long ago (**Moran, 1948**; **Anselin, 1995**) and they were adapted biomedical research much later. There are only sporadic papers using Moran's spatial correlation coefficient (Global Moran's *I*) for biomedical image analysis in order to detect clustering in general (**Guy et al., 2016**) without separating particular clusters. We were interested in localization of clusters that is also possible with the help of a special case of local indicators of spatial association named Local Moran's *I* (**Anselin, 1995**). It can be calculated for each individual pixel of the image:

$$I_i = z_i \sum_j w_{ij} z_j$$

where $z_i$ is the observed pixel intensity value, $z_j$ are the neighboring pixel intensity values weighted by $w_{ij}$. The minimal size of detectable particles is heavily influenced by the size of the weight matrix (**Thompson et al., 2018**). In our case we considered the immediate neighboring pixels ($j$) of each pixel when calculated ($i$) with an equal weight of 1 in first-order matrices. In case of larger matrices, the weights were assigned according to a Gaussian distribution. This coefficient ($I$) gives an estimate of the homogeneity of the spatial distribution of pixels with similar gray values. Moran's *I* is close to zero if the distribution is random, whereas it has a value around −1 if the distribution is uniform and scattered, and +1 if the distribution is uniform and clustered. The result can be statistically tested by a permutation test, where the data points (pixels) are randomly redistributed and the same coefficient is calculated for each permutation. Then a pseudosignificance is computed as $p = (G + 1)/(R + 1)$, where $G$ is the number of instances where the absolute value of the observed *I* is higher than or equal to the absolute value of the original *I*, and $R$ the number of repeats. For example, if none of 99 permutations have a higher *I* value, then the pseudo p-value is 1/100 = 0.01.

Based on the principles above we performed the following step-by-step procedures of Moran's analysis.

1. Normalization of pixel intensity values (*V*)
   $z_i = (V − Vm)/SD$
   where $z_i$ is the normalized intensity value, *V* is the original intensity value, *Vm* is the mean intensity of all pixels, and SD is the standard deviation of intensity of all pixels (**Figure 1a–c**).
2. Creating weights ($w_{ij}$)
   For each pixel, a set of neighboring pixels was defined. These pixels got the weight 1, all others including the actual central pixel got the weight 0. For the present work, we considered the immediate eight neighbors of the pixels, in other words we applied a first-order queen contiguity (**Figure 1d**).
3. Calculating weighted intensity values ($w_{ij}z_j$)After multiplying the weights ($w_{ij}$) with the appropriate normalized pixel intensities ($z_j$), each weighted intensity values belonging to a central pixel were summed ($\sum_j w_{ij}z_j$) (**Figure 1e**).
4. Calculating the lagged gray values that are used for the scatter plots
   $Igv = \sum_j W_{ij}Z_j/n$
   where *n* is the number of neighbors considered (in the corner *n* = 3, on the edge *n* = 5, else *n* = 8, **Figure 1f**).
5. Multiply the lagged gray values with the normalized intensity values gives the Moran's coefficient ($I_i$) for each pixel (**Figure 1g**).
6. Calculation of pseudosignificance (p)

   The pixels of the image were randomly distributed except the central one and the Local Moran's *I* is calculated again. The pseudosignificance level (see above) was calculated for each pixel. The pixels where the p is extreme low, considered to be surrounded by similar pixel values, thus members of homogenous, non-random pixel groups (**Figure 1h**).

Defining clusters The non-random pixel groups can be divided in further sub-groups according to the relation of the pixel intensities of their central ($z_i$) and neighboring (lagged gray value) pixels: high–high, high–low, low–high, and low–low. If both values are high, then this group is regarded a cluster (**Figure 1i, j**).

Previously, this method was used for analyzing geographical distribution of medical data only, but univariate Local Moran's *I* seems to be suitable for delineating clusters of pixels with similar gray values (either high or low) as outliers from the assumed random distribution. The results can be mapped as a graded image and can be used for observer-independent identification of fluorescent clusters. We selected the biologically meaningful spots, which might correspond to pre- or postsynaptic ion channel clusters if they meet the size criteria defined by electron microscopic observations.

## Matrix size

The convolution matrix size defines the number neighboring pixels considered in the convolution step. Traditionally, the size of the matrix is described as the order of the Moran's test. The first-order matrix contains 1 pixel in all directions of a central pixel, that is size of the matrix is 3 × 3, the second-order matrix contains the first-order pixels and their neighbors, therefore the size is 5 × 5, etc. (**Figure 3—figure supplement 2**, orange transparent squares). The matrix size has reportedly a strong effect on Global Moran's *I* results **Thompson et al., 2018**.

Thus, we determine optimal matrix order in our Local Moran's calculation, to find out which provided the best object selection under different noise levels. Each object size (2, 4, 8, 16, 32, and 64 pixels in diameter) was tested with 13 different Moran's order (1–10, 16, 32, and 64) (**Figure 2—figure supplement 2**). The data indicated that independent of noise levels, the best Moran's order is 0.25 times the object size (**Figure 2—figure supplement 1**).

## Manual delineation

Twenty-three volunteers were asked to delineate objects on images. The volunteers did not have any a priori knowledge about the images. The only information that they had is it contains light objects on dark background. Each volunteer received three images in the same order. All three images were the same micrograph of vGluT2 immunostained boutons but with different level of noise (SD of noise was set 160, 80, 0, in this order). Each volunteer must have completed the delineation of all objects of the noisiest image (**Figure 3—figure supplement 1a**) first, before receiving the less noisy and finally the noiseless image (**Figure 3—figure supplement 1b, c**). Human observers drew quite different borders (**Figure 3—figure supplement 1g, i**) but the majority projections (set of pixels selected by the majority of participants, i.e., at least 12 observers) of these object masks resembled to the original structure even in case of the noisiest image (**Figure 3—figure supplement 1d**), but of course it was much clearer in case of the noiseless image (**Figure 3—figure supplement 1f**). Since the biological samples do not have a similar ground truth like the generated objects, we used the areas of such projections of human delineations as a control in the further tests.

## Acknowledgements

This research was supported by the Wellcome Trust (ZN, LA). In addition, LA was supported by an ERC Advanced Grant (FRONTHAL, 742595) and the European Union project RRF-2.3.1-21-2022-00004 within the framework of the Artificial Intelligence National Laboratory and Lendület_2023_90. ZN is the recipient of a Hungarian Academy of Sciences Momentum Grant (Lendület, LP2012-29) and an ERC Advanced Grant (293681). We thank the Light Microscopy Center at Institute of Experimental Medicine for kindly providing microscopy support. Authors would like to express their deepest gratitude to Prof Luc Anselin (Center for Spatial Data Science, University of Chicago) and Dr Szabolcs Káli (Instiute of Experimental Medicine, Budapest) for the valuable discussion about analysis of spatial association, and to Krisztina Faddi for the excellent technical assistance.

## Additional information

### Funding

| Funder | Grant reference number | Author |
|---|---|---|
| European Research Council | FRONTHAL 742595 | Laszlo Acsady |
| European Union | Artificial Intelligence National Laboratory RRF-2.3.1-21-2022-00004 | Laszlo Acsady |
| Wellcome Trust | 10.35802/094513 | Laszlo Acsady Zoltan Nusser |
| European Research Council | 293681 | Zoltan Nusser |
| Hungarian Academy of Sciences | Lendület_2023_90 | Laszlo Acsady |
| Hungarian Academy of Sciences | Lendület LP2012-29 | Zoltan Nusser |

The funders had no role in study design, data collection, and interpretation, or the decision to submit the work for publication. For the purpose of Open Access, the authors have applied a CC BY public copyright license to any Author Accepted Manuscript version arising from this submission.

### Author contributions

Csaba Dávid, Conceptualization, Data curation, Software, Formal analysis, Validation, Investigation, Visualization, Methodology, Writing - original draft, Writing - review and editing; Kristóf Giber, Katalin Kerti-Szigeti, Mihály Köllő, Investigation; Zoltan Nusser, Supervision, Funding acquisition, Methodology, Writing - review and editing; Laszlo Acsady, Conceptualization, Resources, Supervision, Funding acquisition, Methodology, Writing - original draft, Writing - review and editing

### Author ORCIDs

Csaba Dávid ⓘ https://orcid.org/0000-0003-4221-9468
Laszlo Acsady ⓘ https://orcid.org/0000-0002-0679-2980

### Ethics

All animals were bred and kept in the Animal Facility of the Institute of Experimental Medicine, Budapest, Hungary. All procedures with mice were approved by the Animal Welfare Committee of the Institute of Experimental Medicine, Budapest, conformed to guidelines established by the European Community's Council Directive of November 24, 1986 (86/609/EEC). Experiments were authorized by the National Animal Research Authorities of Hungary (PE/EA/877-7/2020).

Reviewer #1 (Public review): https://doi.org/10.7554/eLife.89361.3.sa1
Reviewer #2 (Public review): https://doi.org/10.7554/eLife.89361.3.sa2
Author response https://doi.org/10.7554/eLife.89361.3.sa3

## Additional files

### Supplementary files

• MDAR checklist

### Data availability

The code for the analysis is available online as a user-friendly MATLAB script at: https://github.com/dcsabaCD225/Moran_Matlab/blob/main/moran_local.m (copy archived at *Csaba, 2024*). All source data relevant to figures and charts are included in source data files.

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
