## [Editor Report · eLife assessment]

The manuscript introduces an **important** and innovative non-AI computational method for segmenting noisy grayscale images, with a particular focus on identifying immunostained potassium ion channel clusters. This method significantly enhances accuracy over basic threshold-based techniques while remaining user-friendly and accessible, even for researchers with limited computational backgrounds. The evidence supporting the method's efficacy is **convincing**. Its practical application and ease of use make it a tool that will benefit a wide range of laboratories.

---

## [Referee Report · Reviewer #1 (Public review)]

The manuscript introduces a valuable and innovative non-AI computational method for segmenting noisy grayscale images, with a particular focus on identifying immunostained potassium ion channel clusters.

Strengths:

(1) Applicability and Usability: The method is exceptionally accessible to biologists and researchers without advanced computational expertise. It offers a highly practical alternative to AI-based methods, which often require significant training data and computational resources, making it an excellent choice for a broader range of laboratories.

(2) Proof-of-Concept: The manuscript provides compelling evidence through multiple experiments, showcasing the method's superior performance over traditional threshold-based techniques, particularly in noisy environments. The dual immuno-electron microscopy experiments further reinforce the robustness and effectiveness of this approach.

(3) Clarity and Methodology: The manuscript is exceptionally well-written, with clear and concise descriptions that effectively highlight the method's advantages. The detailed figures and comprehensive references greatly enhance the manuscript's credibility and strongly support the claims made.

Weaknesses:

The manuscript does not include comparisons with more advanced segmentation techniques, particularly those based on artificial intelligence. While the authors have provided a rationale for this decision, including such comparisons could have enriched the discussion and offered additional insights. Additionally, there are some concerns about the computational demands of the method, especially when applied to large-scale or 3D image analysis. Although the authors have shared some computational data, further optimization or practical recommendations would enhance the method's utility. Initially, the manuscript lacked a data and code availability statement, which could have limited the method's accessibility. However, this issue has since been resolved, with the code now being made available to the community. Lastly, while the findings related to Kv4.2 in the thalamus are noteworthy, they might achieve even greater impact if presented in a separate paper. Nevertheless, the authors have chosen to retain these results within the current manuscript to strengthen the overall narrative and relevance.

We appreciate that the authors have provided thorough explanations for their original choices. These justifications offer a clearer understanding of their approach and the reasons behind the presentation of the data.

Conclusion:

The revised manuscript successfully addresses the majority of the reviewers' concerns, presenting a strong case for the proposed segmentation method. The method's ease of use for non-experts in AI, combined with its proven effectiveness in proof-of-concept experiments, positions it as a valuable addition to the field. While the manuscript could benefit from incorporating comparisons with more advanced segmentation methods and offering a more detailed discussion of computational requirements, it remains a robust contribution. The decision to include the Kv4.2 findings within the paper is well-justified by the authors, though these results could potentially have an even greater impact if published separately.

---

## [Referee Report · Reviewer #2 (Public review)]

Summary:

The manuscript by David et al. describes a novel image segmentation method, implementing Local Moran's method, which determines whether the value of a datapoint or a pixel is randomly distributed among all values, in differentiating pixel clusters from the background noise. The study includes several proof-of-concept analyses to validate the power of the new approach, revealing that implementation of Local Moran's method in image segmentation is superior to threshold-based segmentation methods commonly used in analyzing confocal images in neuroanatomical studies.

Strengths:

Several proof-of-concept experiments are performed to confirm the sensitivity and validity of the proposed method. Using composed images with varying levels of background noise and analyzing them in parallel with the Local Moran's or a Threshold-Based Method (TBM), the study is able to compare these approaches directly and reveal their relative power in isolating clustered pixels.

Similarly, dual immuno-electron microscopy was used to test the biological relevance of a colocalization that was revealed by Local Moran's segmentation approach on dual-fluorescent labeled tissue using immuno-markers of the axon terminal and a membrane-protein (Figure 5). The EM revealed that the two markers were present in terminals and their post-synaptic partners, respectively. This is a strong approach to verify the validity of the new approach for determining object-based colocalization in fluorescent microscopy.

The methods section is clear in explaining the rationale and the steps of the new method (however, see the weaknesses section). Figures are appropriate and effective in illustrating the methods and the results of the study. The writing is clear; the references are appropriate and useful.

Weaknesses:

While the steps of the mathematical calculations to implement Local Moran's principles for analyzing high-resolution images are clearly written, the manuscript currently does not provide a computation tool that could facilitate easy implementation of the method by other researchers. Without a user-friendly tool, such as an ImageJ plugin or a code, the use of the method developed by David et al by other investigators may remain limited.

This weakness is eliminated in the revision, which now provides the approach as a Matlab tool.

---

## [Author Response]

The following is the authors’ response to the original reviews.

**Public Reviews:**

**Reviewer #1 (Public Review):**
The study describes a new computational method for unsupervised (i.e., non-artificial intelligence) segmentation of objects in grayscale images that contain substantial noise, to differentiate object, no object, and noise. Such a problem is essential in biology because they are commonly confronted in the analysis of microscope images of biological samples and recently have been resolved by artificial intelligence, especially by deep neural networks. However, training artificial intelligence for specific sample images is a difficult task and not every biological laboratory can handle it. Therefore, the proposed method is particularly appealing to laboratories with little computational background. The method was shown to achieve better performance than a threshold-based method for artificial and natural test images. To demonstrate the usability, the authors applied the method to high-power confocal images of the thalamus for the identification and quantification of immunostained potassium ion channel clusters formed in the proximity of large axons in the thalamic neuropil and verified the results in comparison to electron micrographs.Strengths:The authors claim that the proposed method has higher pixel-wise accuracy than the threshold-based method when applied to gray-scale images with substantial noises.Since the method does not use artificial intelligence, training and testing are not necessary, which would be appealing to biologists who are not familiar with machine learning technology.The method does not require extensive tuning of adjustable parameters (trying different values of "Moran's order") given that the size of the object in question can be estimated in advance.

We appreciate the positive assessment of our approach.

Weaknesses:It is understood that the strength of the method is that it does not depend on artificial intelligence and therefore the authors wanted to compare the performance with another non-AI method (i.e. the threshold-based method; TBM). However, the TBM used in this work seems too naive to be fairly compared to the expensive computation of "Moran's I" used for the proposed method. To provide convincing evidence that the proposed method advances object segmentation technology and can be used practically in various fields, it should be compared to other advanced methods, including AI-based ones, as well.

Protein localization studies revealed that protein distributions are frequently inhomogeneous in a cell. This is very common in neurons which are highly polarized cell types with distinct axo-somato-dendritic functions. Moreover, due to the nature of the cell-to-cell interactions among neurons (e.g. electrical and chemical synapses) the cell membrane includes highly variable microdomains with unique protein assemblies (i.e. clusters). Protein clusters are defined as membrane segments with higher protein densities compared to neighboring membrane regions. However, protein density can continuously change between “clusters” and “non-clusters”. As a consequence, differentiating proteins involved vs not involved in clusters is a challenging task. Indeed, our analysis showed that the boundaries of protein clusters varied remarkably when 23 human experts delineated them.

Despite the fact the protein clusters can only be vaguely defined numerous studies have demonstrated the functional relevance of inhomogeneous protein distribution. Thus, there is a high relevance and need for an observer independent, “operative” segmentation method that can be accomplished and compared among different conditions and specimens. The strength of the Moran’s I analysis we propose here, as pointed out by our reviewers and editors, is that it can extract the relevant signals from an image generated in different, often noisy condition using a simple algorithm that allows quantitative characterization and identification of changes in many biological and non-biological samples.

In AI based analysis the ground truth is known by an observer and using a large training set AI learns to extract the relevant information for image segmentation. As outlined above the “ground truth”, however, cannot be unequivocally defined for protein clusters. There is no doubt, that with sufficient resource investment there would be an AI based analysis of the same problem. In our view, however, in an average laboratory setting generating a training set using hundreds of images examined by many experts may not be plausible. Moreover, generalization of one training set to another set of cluster, resistance to noise or different levels of background could also not be guaranteed.

This method was claimed to be better than the TBM when the noise level was high. Related to the above, TBMs can be used in association with various denoising methods as a preprocess. It is questionable whether the claim is still valid when compared to the methods with adequate complexity used together with denoising. Consider for example, Weigert et al. (2018) https://doi.org/10.1038/s41592-018-0216-7; or Lehtinen et al (2018) https://doi.org/10.48550/arXiv.1803.04189.

In Weigert et al. AI was trained with high-quality images of the same object obtained with extreme photon exposure in confocal microscope. As delineated above without training AI systems cannot be used for such purposes. The Lehtinen paper is unfortunately no longer available at this doi.

We must emphasize that in our work we did not intend to compare the image segmentation method based on local Moran’s I with all other available segmentation techniques. Rather we wanted to demonstrate a straightforward method of grouping pixels with similar intensities and in spatial proximity which does not require a priori knowledge of the objects. We used TBM to benchmark the method. We agree that with more advanced TBM methods the difference between Moran’s and TBM might have been smaller. The critical component here is, however, that even with most advanced TBM an artificial threshold is needed to be defined. The optimal threshold may change from sample to sample depending on the experimental conditions which makes quantification questionable. Moran’s method overcomes this problem and allows more objective segmentation of images even if the exact conditions (background labeling, noise, intensity etc) are not identical among the samples.

The computational complexity of the method, determined by the convolution matrix size (Moran's order), linearly increases as the object size increases (Fig. S2b). Given that the convolution must be run separately for each pixel, the computation seems quite demanding for scale-up, e.g. when the method is applied for 3D image volumes. It will be helpful if the requirement for computer resources and time is provided.

Here we provide the required data concerning the hardware and the computational time:

Hardware used for performing the analysis:

Intel(R) Xeon(R) Silver 4112 CPU @ 2.60GHz, 2594 Mhz, 4 kernel CPU, 64GB RAM, NVIDIA GeForce GTX 1080 graphic card.

MATLAB R2021b software was used for implementation.

**Author response table.**

**Author response table 1. sa3table1:** Computation times.

	Computation time (s) of images with size of
Moran’s order		64	128	256	512	1024	2048
1	1.82	2.3	2.7	6.18	17.79	63.91
2	2.21	2.22	1.91	6.1	18.08	64.84
4	2.04	3.01	2.65	5.41	18.9	66.53
8	2.15	2.39	2.67	6.58	20.06	72.54
16	2.06	2.24	3.18	7.68	25.57	106.71

**Reviewer #2 (Public Review):**
Summary:The manuscript by David et al. describes a novel image segmentation method, implementing Local Moran's method, which determines whether the value of a datapoint or a pixel is randomly distributed among all values, in differentiating pixel clusters from the background noise. The study includes several proof-of-concept analyses to validate the power of the new approach, revealing that implementation of Local Moran's method in image segmentation is superior to threshold-based segmentation methods commonly used in analyzing confocal images in neuroanatomical studies.Strengths:Several proof-of-concept experiments are performed to confirm the sensitivity and validity of the proposed method. Using composed images with varying levels of background noise and analyzing them in parallel with the Local Moran's or a Threshold-Based Method (TBM), the study is able to compare these approaches directly and reveal their relative power in isolating clustered pixels.Similarly, dual immuno-electron microscopy was used to test the biological relevance of a colocalization that was revealed by Local Moran's segmentation approach on dual-fluorescent labeled tissue using immuno-markers of the axon terminal and a membrane-protein (Figure 5). The EM revealed that the two markers were present in terminals and their post-synaptic partners, respectively. This is a strong approach to verify the validity of the new approach for determining object-based colocalization in fluorescent microscopy.The methods section is clear in explaining the rationale and the steps of the new method (however, see the weaknesses section). Figures are appropriate and effective in illustrating the methods and the results of the study. The writing is clear; the references are appropriate and useful.

We are grateful for the constructive assessment of our results.

Weaknesses:While the steps of the mathematical calculations to implement Local Moran's principles for analyzing high-resolution images are clearly written, the manuscript currently does not provide a computation tool that could facilitate easy implementation of the method by other researchers. Without a user-friendly tool, such as an ImageJ plugin or a code, the use of the method developed by David et al by other investigators may remain limited.

The code for the analysis is now available online as a user-friendly MATLAB script at: https://github.com/dcsabaCD225/Moran_Matlab/blob/main/moran_local.m

**Recommendations for the authors:**
Summary of reviews:Both reviewers acknowledge the potential significance and practicality of the newly proposed image segmentation method. This method uses Local Moran's principles, offering an advantage over traditional intensity thresholding approaches by providing more sensitivity, particularly in reducing background noise and preserving biologically relevant pixels.Strengths Highlighted:• The proposed method can provide more accurate results, especially for grayscale images with significant noise.• The method is not dependent on artificial intelligence, making it appealing for researchers with minimal computational background.• The approach can operate without the need for extensive tuning, given that the size of the object is known.• Several proof-of-concept experiments were carried out, revealing the effectiveness of the method in comparison with the threshold-based segmentation methods.• The manuscript is clear in terms of methodology, and the results are supported by effective illustrations and references.Weaknesses Noted:• The study lacked a comparative analysis with advanced segmentation methods, especially those that employ artificial intelligence.

See our response above to the same question of Reviewer 1.

• There are concerns about computational complexity, especially when dealing with larger data sets or 3D image volumes.

See our response about the calculations of computation times above to the similar question of Reviewer 1.

• Both reviewers noted the absence of a data/code availability statement in the manuscript, which might restrict the method's adoption by other researchers.

The code availability is provided now.

• Reviewer 2 suggested that some results, particularly related to Kv4.2 in the thalamus, might be better presented in a separate study due to their significance.

We thank our reviewers for this suggestion. We carefully evaluated the pros and cons of publishing the Kv4.2 data separately. We finally decided to keep the segmentation and experimental data together due to the following reason. We believe that the ultrastructural localization provides strong experimental proof for the relevance of our novel segmentation method. In order to make the potassium channel data more visible we added a subsentence to the title. In this manner we think scientist interested in the imaging method as well as the neurobiology will be both find and cite the paper. The novel title reads now:

“An image segmentation method based on the spatial correlation coefficient of Local Moran’s I - identification of A-type potassium channel clusters in the thalamus.”

**Reviewer Recommendations:**
(1) Provide details about the data and program code availability.

See our response above

(2) Offer practical recommendations and provide clarity on software packages and coding for the proposed method to enhance its adoption.

Done.

(3) Consider presenting the findings about Kv4.2 in the thalamus separately as they hold significant importance on their own.

See our response above

Given the reviews, the proposed image segmentation method presents a promising advancement in the domain of image analysis. The technique offers tangible benefits, especially for researchers dealing with biological microscopy data. However, for this method to see a broader application, it's imperative to provide clearer practical guidance and make data or code easily accessible. Additionally, while the findings regarding Kv4.2 in the thalamus are intriguing, they might achieve more impact if detailed in a dedicated paper.
**Reviewer #1 (Recommendations For The Authors):**
The availability of data or program code was not stated in the manuscript.
**Reviewer #2 (Recommendations For The Authors):**
(1) While the principles of the method are explained clearly in a step-by-step fashion in the Methods section, the practical aspects of running sequential computations over a large matrix of pixel values are not well described. It would be very useful if the authors could provide recommendations on how to set the data structure and clarify which software and programming package for Local Moran's analysis they used. In addition, providing the code for the sequential implementation described in the Methods section would facilitate the adoption of the method by other researchers, and thus, the impact of the study. Currently, there is no data or code availability statement included in the manuscript.

See our response above.

(2) Figure 4 illustrates an experiment in which transmission electron microscopy and freeze-fracture replica labeling approaches were used to demonstrate that a potassium channel marker, Kv4.2 was selective to synapses forming on larger caliber dendrites in the thalamus. As impressive as the EM approaches utilized in this figure are, the results of this experiment have a somewhat tangential bearing on the segmentation method that is the focus of this study. In fact, the experiments illustrated in Figure 5, dual immuno-EM, are more than sufficient to confirm what the dual-confocal imaging coupled with Local Moran's segmentation analysis reveals. Furthermore, the author's findings about the localization and selectivity of Kv4.2 in the thalamus are too important and exciting to bury in a paper focusing on the methodology. Those results may have a wider impact if they are presented and discussed in a separate experimental paper.

See our response above